# Aggregation pheromone 4-vinylanisole promotes the synchrony of sexual maturation in female locusts

**Dafeng Chen[1†], Li Hou[1,2†], Jianing Wei[1], Siyuan Guo[1], Weichan Cui[1], Pengcheng Yang[3], Le Kang[1,2,3]\*, Xianhui Wang[1,2]\***

[1]State Key Laboratory of Integrated Management of Pest Insects and Rodents, Institute of Zoology, Chinese Academy of Sciences, Beijing, China; [2]CAS Center for Excellence in Biotic Interactions, University of Chinese Academy of Sciences, Beijing, China; [3]Beijing Institutes of Life Sciences, Chinese Academy of Sciences, Beijing, China

**\*For correspondence:**
lkang@ioz.ac.cn (LK);
wangxh@ioz.ac.cn (XW)

[†]These authors contributed equally to this work

**Competing interest:** The authors declare that no competing interests exist.

**Abstract** Reproductive synchrony generally occurs in various group-living animals. However, the underlying mechanisms remain largely unexplored. The migratory locust, *Locusta migratoria*, a worldwide agricultural pest species, displays synchronous maturation and oviposition when forms huge swarm. The reproductive synchrony among group members is critical for the maintenance of locust swarms and population density of next generation. Here, we showed that gregarious female locusts displayed more synchronous sexual maturation and oviposition than solitarious females and olfactory deficiency mutants. Only the presence of gregarious male adults can stimulate sexual maturation synchrony of female adults. Of the volatiles emitted abundantly by gregarious male adults, the aggregation pheromone, 4-vinylanisole, was identified to play key role in inducing female sexual maturation synchrony. This maturation-accelerating effect of 4-vinylanisole disappeared in the females of Or35[-/-] lines, the mutants of 4-vinylanisole receptor. Interestingly, 4-vinylanisole displayed a time window action by which mainly accelerates oocyte maturation of young females aged at middle developmental stages (3–4 days post adult eclosion). We further revealed that juvenile hormone/vitellogenin pathway mediated female sexual maturation triggered by 4-vinylanisole. Our results highlight a 'catch-up' strategy by which gregarious females synchronize their oocyte maturation and oviposition by time-dependent endocrinal response to 4-vinylanisole, and provide insight into reproductive synchrony induced by olfactory signal released by heterosexual conspecifics in a given group.

## Editor's evaluation

This study describes an important aspect of the devastating swarming behaviour of locusts – how gregarious female locusts might synchronise oviposition. It uncovers a role for olfactory signaling. The authors find that the aggregation pheromone 4-vinylanisole released by gregarious males is instrumental in synchronization of female sexual maturation. This study will be useful for the understanding of swarming behaviour in locusts, and it will also interest those who work on behaviour and its modulation.

## Introduction

Reproductive synchrony, characterized by a pronounced temporal clustering of births, estrus, or mating, widely occurs in the animal kingdom, especially in group-living species (***Ims, 1990***). Several

**eLife digest** Since 2019, a plague of flying insects known as migratory locusts has been causing extensive damage to crops in East Africa. Migratory locusts sometimes live a solitary lifestyle but, if environmental conditions allow, they form large groups containing millions of individuals known as swarms that are responsible for causing locust plagues.Locusts are able to maintain such large swarms because they can aggregate and synchronize.

When they live in swarms, individual locusts produce odors that are sensed by other individuals in the group. For example, an aggregation pheromone, called 4-vinylanisole, is known to help keep large groups of locusts together. However, it is less clear how odors synchronize the reproductive cycles of the females in a swarm so that they are ready to mate with males and lay their eggs at the same time.

To address this question, Chen et al. examined when female locusts reached sexual maturity after they were exposed to odors produced by other locusts living alone or in groups. The experiments found that only 4-vinylanisole, which was abundantly released by adult male locusts living in groups, stimulated female locusts to reach sexual maturity at the same time. This odor increased the levels of a hormone known as juvenile hormone in less-developed females to help them reach sexual maturity sooner.

These findings demonstrate that when migratory locusts are living in swarms, male locusts promote the female locusts to reach sexual maturity at the same time by promoting less-developed females to 'catch up' with other females in the group. A next step will be to investigate the neural and molecular mechanisms underlying the 'catch up' effect induced by 4-vinylanisole.

prominent cases are best known for their extreme manifestations, for example, sea turtle oviposition, firefly flashing, and fish spawning, involving a mass of individuals with the same reproductive state at certain time windows (*Buck and Buck, 1968*; *Harrison et al., 1984*; *Kelly and Sork, 2002*). Reproductive synchrony may offer adaptive advantages for group-living species, such as predation swamping and inbreeding avoidance (*Janzen, 1971*). Therefore, understanding how reproductive cycle is synchronized among individuals would provide insight to the biological flexibility in group-living animals.

Reproductive synchrony is a complex process that requires the integration of extra- and endo-signals to coordinate the timing of reproductive cycles between individuals in a group (*Kobayashi et al., 2002*; *Dey et al., 2015*). In fact, intra-group variation in developmental status can be induced by many factors, including different nutrition, temperature, and order of eclosion (*Ward and Webster, 2016*), which essentially makes synchronous reproduction between all members an apparent improbability. Social interaction is considerably critical for triggering reproductive synchrony of individuals in group-living species (*French and Stribley, 1985*; *Ims and Steen, 1990*; *Jovani and Grimm, 2008*). A well-known example is the Whitten effect which is induced by the presence of males in rodents, ewes, and monkey (*Vandenbergh, 1967*; *Cahill et al., 1974*; *Gattermann et al., 2002*). Various kinds of signals, odors, touch, or voice can act as social clues to underpin synchronization with reproduction (*Rekwot et al., 2001*; *Kobayashi et al., 2002*; *Noguera and Velando, 2019*). Endo-signals, such as hormone release, gene expression, and epigenetic modification, have also been suggested to be involved in these interaction processes (*Engel et al., 2016*; *Noguera and Velando, 2019*). However, the mechanisms by which social cue/hormone interaction synchronizes the reproductive cycles of individuals within local breeding groups remain largely unknown.

Locusts often form large swarms with hundreds to thousands of individuals, regarded as one of the most extraordinary examples of coordinated behavior (*Ariel and Ayali, 2015*; *Buhl and Rogers, 2016*). Depending on population density, locusts display striking phenotypic plasticity, with a cryptic solitarious phase and an active gregarious phase (*Wang and Kang, 2014*). Gregarious locusts, compared to solitarious conspecifics, show much higher synchrony in physiological and behavioral events, such as egg hatching and sexual maturation, as well as synchronous feeding and marching behaviors (*Norris, 1954*; *Uvarov, 1977*). Reproductive synchrony in gregarious locusts provides benefits for individuals in several aspects, such as more favorable microenvironment, lower risk of predation, efficiently forging, as well as more encounters with mates, therefore

ensures high-density conditions for the next generation, and is essential for maintenance of locust swarm (*Beekman et al., 2008*, *Maeno et al., 2021*). Some sort of vibratory stimulus, maternal microRNAs, and SNARE protein play important roles in the egg-hatching synchrony of gregarious locusts (*Chen et al., 2015*; *He et al., 2016*; *Nishide and Tanaka, 2016*). It has been revealed that the presence of mature male adults has effectively accelerating effects on synchrony of sexual maturation of immature male and female conspecifics in two locust species, *Schistocerca gregaria* and *Locusta migratoria* (*Norris, 1952*; *Loher, 1997*; *Guo and Xia, 1964*; *Norris and Richards, 1964*). The accelerating effects of several prominent volatiles released by gregarious mature males in male maturation have been examined in the desert locust. Four volatile pheromones (benzaldehyde, veratrole, phenylacetonitrile [PAN], and 4-vinylveratrole) have significantly stimulatory effects on sexual maturation of male adults, with PAN having the most pronounced effect (*Mahamat et al., 1993*; *Assad et al., 1997*). However, how conspecific interactions affect female sexual maturation remain unclear and the pheromones those contribute to maturation synchrony of females have not been determined so far.

In this study, we investigated mechanisms underlying sexual maturation synchrony of female adults in the migratory locust by comparing phase-related maturation patterns of females using multidisciplinary studies, including physiology, chemical ecology, genomics, and gene manipulation. Unexpectedly, we found that aggregation pheromone, 4-vinylanisole, induced sexual maturation synchrony of female adult locusts. Our results highlight a parsimonious role of olfactory cues in the formation of locust swarms by triggering aggregation behavior and sexual maturation synchrony.

## Results
### Olfactory signals from gregarious male adults trigger the synchrony of female sexual maturation in locusts

We first investigated whether there was a difference of reproduction synchrony between gregarious and solitarious female locusts by determining the distribution of first oviposition date. The curve of the first oviposition time of gregarious females was much narrower than that of solitarious females (60% decrease in the standard deviation [SD], *Figure 1A*), implying that the first reproductive cycle was more consistent among gregarious female individuals. As an essential premise of egg laying, sexual maturation states, indicated by the length of terminal oocyte relative to the final mature size, were then measured. The development of terminal oocyte increased with ages of female adults of both phases. Gregarious female adults displayed more uniform and rapid patterns than that of solitarious females after 4 days post adult eclosion (PAE 4 days) (*Figure 1B* and *Figure 1—figure supplement 1*). These results indicate that gregarious female adults display significant sexual maturation synchrony and higher maturation rate.

We next investigated whether conspecific interactions can induce sexual maturation synchrony of female adults (*Figure 1—figure supplement 2A* and B). We found that the maturation synchrony of terminal oocytes of females was significantly retarded by the removal of male adults in gregarious phase (*Figure 1C*). However, raised with either solitarious female or male did not affect sexual maturation of solitarious females (*Figure 1D*). The exposure of odor blends from gregarious male adults significantly advanced the maturation synchrony of gregarious females and solitarious females, whereas no effects were observed when exposed to the background air, female odors, or odors from solitarious males (*Figure 1E and F* and *Figure 1—figure supplement 2C, D*).

To further explore the roles of olfactory cues in females' sexual maturation process, we examined the performance of loss-of-function mutants of olfactory receptor co-receptor gene (Orco$^{-/-}$) established by CRISPR/Cas9, which display significant olfactory deficiency (*Li et al., 2016*). The best-fit normal curve of the first oviposition date was much wider in Orco$^{-/-}$ females than in wildtype (WT) females, when they were reared together with gregarious males (with 63% increase in the SD, *Figure 1G*). When reared together with gregarious male adults or exposure to their odor blends, the sexual maturation of Orco$^{-/-}$ females was less synchronous than that of WT females (*Figure 1H, I* and *Figure 1—figure supplement 2E, F*). Thus, olfactory signals from gregarious male adults are essential for triggering the synchrony of female sexual maturation in migratory locusts.

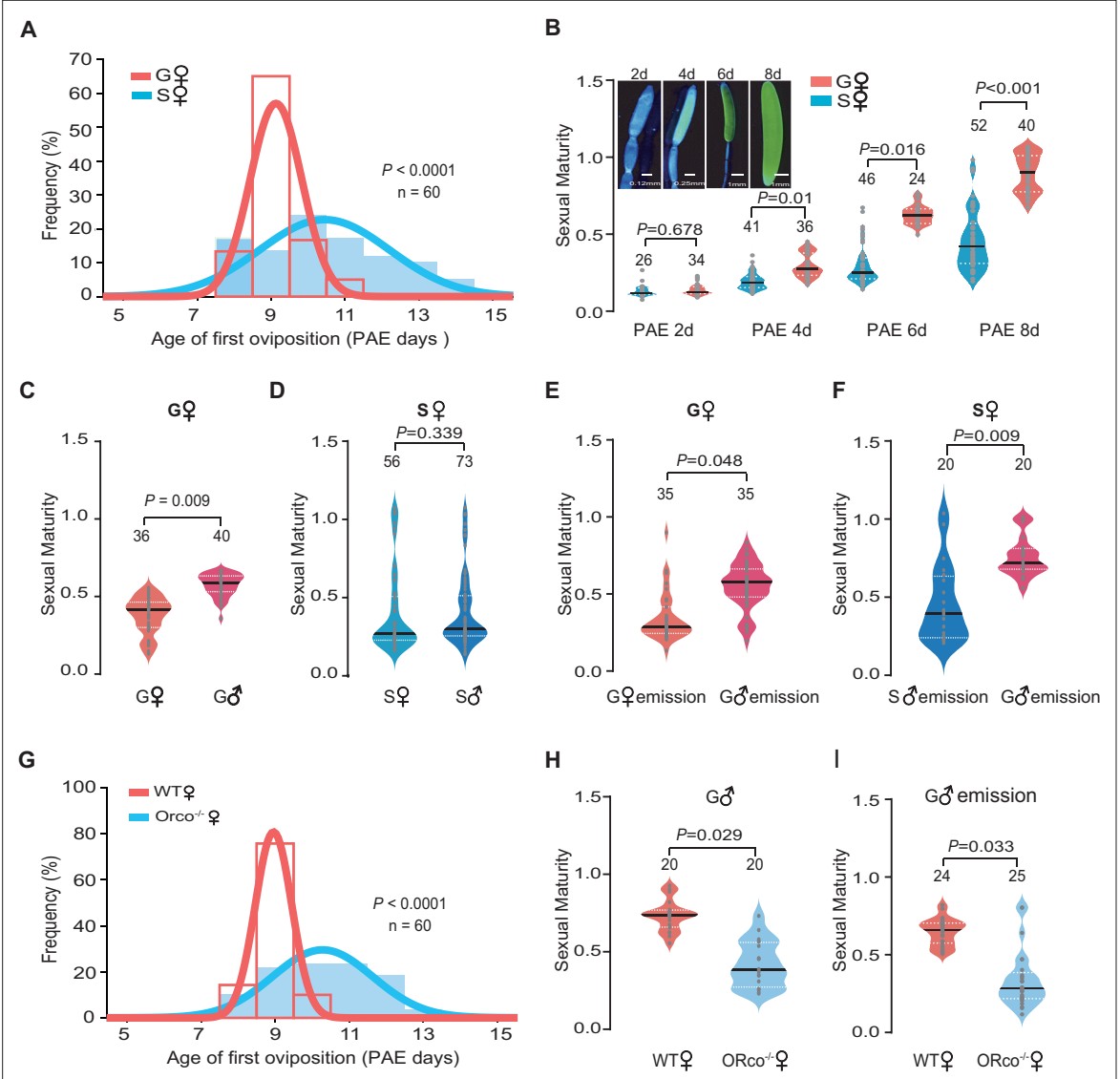

**Figure 1.** Olfactory signals from gregarious male adults trigger the synchrony of female sexual maturation in locusts. (**A**) Distribution of the first oviposition time of gregarious and solitarious phases. The first oviposition date was recorded after 6 days post adult eclosion (PAE 6 days) when individuals began to mate. To ensure the consistency of mating time in gregarious and solitarious locusts, females that did not successfully mate with 24 hr after pairing were excluded. For gregarious locusts, females were individually marked, and their first oviposition times were recorded by collecting egg pods every 4 hr per day after mating. Females those laid new eggs could be easily distinguished by much thinner abdomen with white foam around ovipositor. Ages of first oviposition were indicated by days post eclosion. (**B**) The maturity of gregarious and solitarious females from PAE 2 to 8 days. The sexual maturity was presented as the length of terminal oocyte relative to the final mature size. (**C**) The maturity of gregarious females reared with gregarious males or females, separately. (**D**) The maturity of solitarious females reared with solitarious males or females, separately. (**E**) The maturity of gregarious females stimulated by volatiles released from gregarious males or females. (**F**) The maturity of solitarious females stimulated by volatiles released from gregarious or solitarious males. (**G**) Distribution of the first oviposition time in wild-type (WT) females and Orco female mutants (Orco⁻/⁻). (**H**) The maturity of WT females and Orco⁻/⁻ females reared with gregarious males. (**I**) The maturity of WT females and Orco⁻/⁻ females stimulated by volatiles released from gregarious males. Only virgin females were used in all experiments refer to sexual maturation examination. Dark lines in violin plots indicate median value. White dotted lines indicate upper and lower quartiles, respectively. Consistency analysis was analyzed using Levene's test. The number of biological replicates and p values were shown in the figures.

The online version of this article includes the following source data and figure supplement(s) for figure 1:

**Source data 1.** Raw data for first oviposition time and sexual maturity of gregarious, solitarious, and Orco⁻/⁻ female adults.

**Figure supplement 1.** Comparison of sexual maturation rate of gregarious and solitarious females.

**Figure supplement 1—source data 1.** Raw data for maturation rate of females between gregarious and solitarious phases.

**Figure supplement 2.** Schematic diagram of the stimulation experiments for females of the migratory locust.

## 4-Vinylanisole abundantly released by gregarious male adults mediates sexual maturation synchrony of female locusts

To identify key active compounds that can promote sexual maturation synchrony of female locusts, we compared the volatile emission dynamics of gregarious male adults, gregarious female adults, and solitarious male adults from PAE 1 to 8 days. In total, 14 chemicals were identified in the volatiles released by male adults (*Figure 2A*). After PAE 4 days, only five compounds displayed considerably higher abundance in gregarious male adults, compared to that released by gregarious female adults and solitarious male adults, which showed no accelerating effects on female sexual maturation synchrony. Thus, these five kinds of gregarious male adult-abundant volatiles, including PAN, guaicol, 4-vinylanisole (4-VA), vertrole, and anisole (*Figure 2A* and *Figure 2—figure supplement 1*), might serve as candidate pheromones for female maturation acceleration.

We exposed gregarious young female locusts for 6 days after fledging to different synthetic blends of those five compounds (PAN, guaicol, 4-VA, vertrole, and anisole). The full blend of five components was effective in promoting the synchrony of oocyte development. Only the omission of 4-VA, but not other four compounds, from the full blend lost the accelerating effects on sexual maturation synchrony of gregarious females (*Figure 2B*). Moreover, the exposure to 4-VA can induce similar effects on female sexual maturation synchrony to the full blend (*Figure 2C*). In addition, the accelerating effects of 4-VA on maturation synchrony displayed a dose-threshold pattern, with an effective concentration more than 100 ng (*Figure 2D* and *Figure 2—figure supplement 2*). We further examined the performance of Or35[-/-] females that cannot sense 4-VA (*Guo et al., 2020*). Compared to WT females, the best-fit normal curve of the first oviposition date of Or35[-/-] females was much wider (52% increase in the SD, *Figure 2E*). The sexual maturation in Or35[-/-] females was more uneven than that of WT females when they were reared together with gregarious males (*Figure 2F*) or exposed to the odors of gregarious males (*Figure 2—figure supplement 3*). Moreover, the synchronous effects of 4-VA completely disappeared in Or35[-/-] females (*Figure 2G*).

## 4-VA action needs a critical time window on sexual maturation synchrony in young females

In fact, intra-group variation generally exists in the maturation period of female locusts due to differences in nymph experience, nutrition, and fledging time (*Uvarov, 1977*). We hypothesized that there is a differential effect of 4-VA on the maturation rate for female individuals with distinct developmental statuses to achieve maturation synchrony. To test this, we determined the accelerating effects of 4-VA on young females at three different ages after fledging: PAE 1–2 days, 3–4 days, and 5–6 days, respectively. We found that female maturation synchrony was significantly enhanced only when young females were treated by 4-VA at PAE 3–4 days, while did not change at PAE 1–2 days or PAE 5–6 days, indicating the time window of 4-VA action on sexual maturation of females (*Figure 3A*). Moreover, we compared the effects of gregarious males with different ages on female maturation. The maturation synchrony of females was significantly enhanced by gregarious males aged at PAE 3–4 days and PAE 5–6 days (*Figure 3—figure supplement 1*), which could release more 4-VA (*Figure 2—figure supplement 1*). By contrast, rearing together with the fifth instar of gregarious males and male adults aged at PAE 1–2 days did not significantly affect the maturation synchrony of gregarious female adults (*Figure 3—figure supplement 1*).

Gene expression profiles in fat body tissue have been demonstrated to correlate tightly with the sexual maturation of female locusts (*Guo et al., 2014*). Therefore, we further evaluated the time window effects of 4-VA on female sexual maturation at the transcriptomic level. Through RNA-seq, we verified that the gene expression profiles of fat body displayed more remarkable changes in female adults exposed to 4-VA at PAE 3–4 days (1700 differentially expressed genes [DEGs]) than those at PAE 1–2 days (505 DEGs) and at PAE 5–6 days (582 DEGs) (*Figure 3—figure supplement 2*). Meanwhile, Kyoto encyclopedia of genes and genomes enrichment analysis showed that there were more signal pathways affected by 4-VA treatment at PAE 3–4 days than PAE 1–2 days and PAE 5–6 days. Notably, genes related to energy metabolism, such as retinol metabolism, glycerolipid metabolism, pyruvate metabolism, as well as fatty acid biosynthesis, which play essential roles in ovary development, were significantly activated by 4-VA treatment at PAE 3–4 days (*Figure 3—figure supplement 2*). Thus, PAE 3–4 days should be a critical time window for 4-VA-induced acceleration of female sexual maturation.

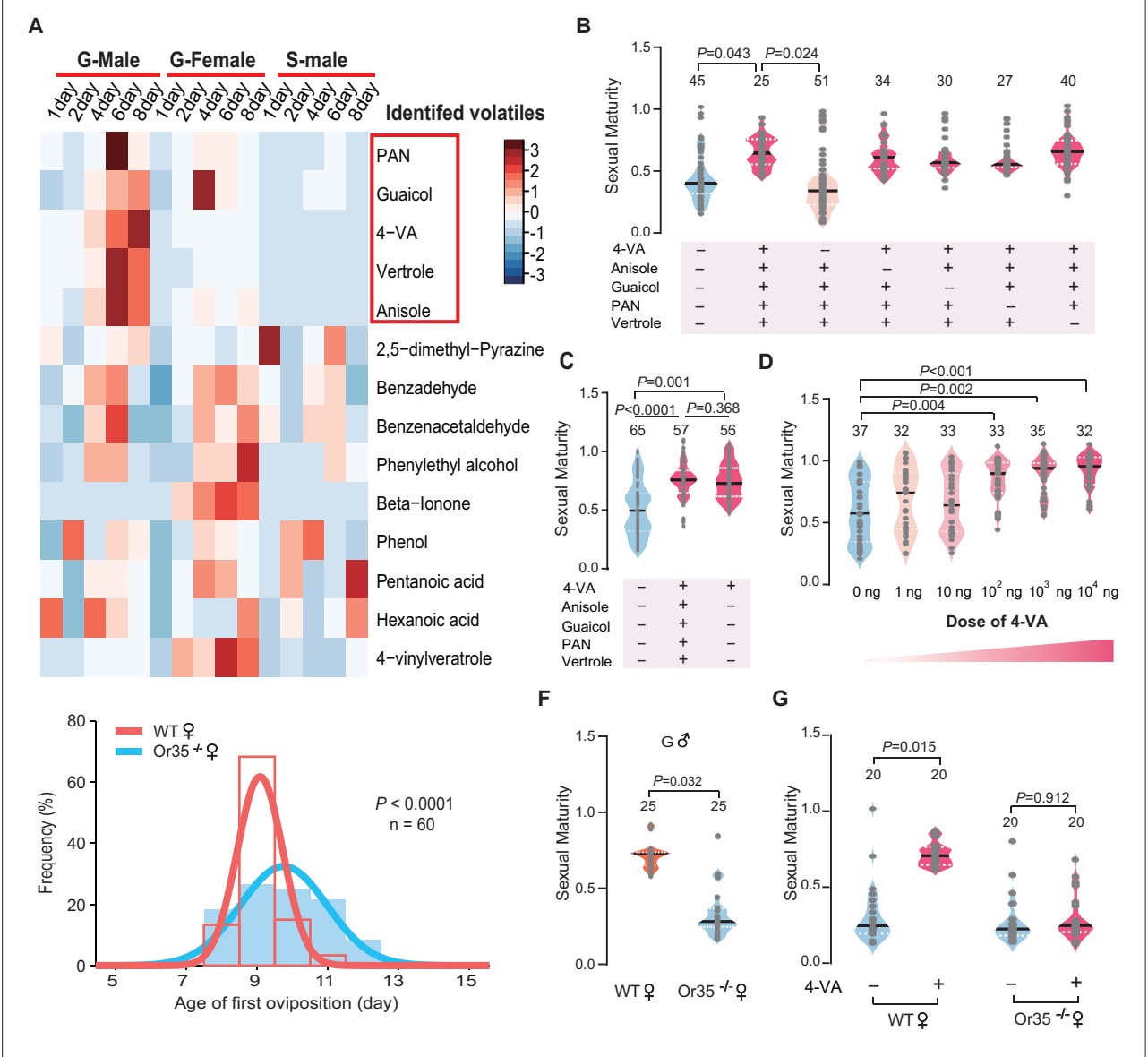

**Figure 2.** 4-Vinylanisole abundantly released by gregarious male adults promotes sexual maturation synchrony of female locusts. (**A**) Dynamic changes in volatiles released from gregarious male adults, gregarious female adults, and solitarious male adults from post adult eclosion (PAE) 1 to 8 days. (**B**) The maturity of gregarious females stimulated with different volatile mixtures containing gregarious male-abundant compounds. Ten gregarious virgin females were stimulated by the mixed odor blend (phenylacetonitrile [PAN], guaicol, 4-vinylanisole [4-VA], vertrole, and anisole) or paraffin oil from PAE 1 to 6 days. (**C**) The maturity of gregarious females treated with five kinds of gregarious male-abundant volatiles or 4-VA alone. (**D**) Dosage effects on the maturity of gregarious females after 4-VA stimulation. (**E**) Distribution of the first oviposition time in wild-type (WT) and Or35[-/-] females. (**F**) The maturity of WT and Or35[-/-] females reared with gregarious male adults. (**G**) The maturity of WT and Or35[-/-] females with or without 4-VA stimulation. Dark lines in violin plot indicate median value. White dotted lines indicate upper and lower quartile, respectively. Consistency analysis was analyzed using Levene's test. The number of biological replicates and p values were shown in the figures.

The online version of this article includes the following source data and figure supplement(s) for figure 2:

**Source data 1.** Raw data for volatile contents in adults and sexual maturity of females.

**Figure supplement 1.** Releasing dynamics of 14 identified volatiles in gregarious male adults, gregarious female adults, and solitarious male adults from post adult eclosion (PAE) 1 to 8 days.

**Figure supplement 2.** Dose-dependent effects of 4-vinylanisole (4-VA) on female maturation rate.

**Figure supplement 2—source data 1.** Raw data for maturation rate of females treated by 4-vinylanisole (4-VA).

*Figure 2 continued on next page*

*Figure 2 continued*

**Figure supplement 3.** The sexual maturity of wild-type (WT) females and Or35⁻/⁻ females after stimulation by volatiles released from gregarious males (n = 20, Levene's test, p = 0.02; Student's t-test, ***p < 0.001).

**Figure supplement 3—source data 1.** Raw data for sexual maturity of wild-type (WT) females and Or35⁻/⁻ females stimulated by volatiles released from gregarious males.

## JH/Vg signaling pathway mediates the accelerating effect of 4-VA on sexual maturation synchrony in young females

To explore the regulatory mechanism underlying the time window effects of 4-VA on the sexual maturation synchrony of female locusts, we examined the performance of major signaling pathways involved in the sexual maturation of female locusts. First, we determined whether females display time-dependent electrophysiological response to 4-VA by performing electroantennography (EAG) and single sensilla response (SSR) experiments. We found that 4-VA-induced EAG and SSRs of female adults displayed obvious dose-dependent effects (*Figure 3B* and *Figure 3—figure supplement 3*). However, there was no difference of EAG and SSRs of females among the ages of PAE 2 days, PAE 4 days, and PAE 6 days, although LmigOr35 expression levels were dynamic during ovary development (*Figure 3B* and *Figure 3—figure supplements 3 and 4*). These results suggested that peripheral olfactory perception may not be involved in the time window effects of 4-VA.

We then compared gene expression profiles of two main neuroendocrinal tissues, the brain and corpus cardiacum-corpora allatum (CC-CA) complex, between the controls and 4-VA-exposed females at PAE 3–4 days. Notably, gene expression profiles in CC-CA significantly changed upon 4-VA treatment, with 290 DEGs, much more than that in the brain (89 DEGs) (*Figure 3C* and *Figure 3—figure supplement 5*), implying the molecular and physiological activities in CC-CA might be remarkably affected by 4-VA stimuli. Moreover, a series of DEGs of CC-CA involved in juvenile hormone (JH) metabolism were enriched. There was significantly higher expression of genes related to JH synthesis but lower expression of genes associated with JH degradation (*Figure 3C and D* and *Figure 3—figure supplement 6* and *Supplementary file 1*). These results indicated a potential role of JH signaling pathway in mediating the effects of 4-VA at PAE 3–4 days.

We therefore tested whether 4-VA exposure can affect the hemolymph JH titer in immature females. As expected, the JH titer was significantly elicited by 4-VA exposure in females aged at PAE 3–4 days, rather than PAE 1–2 days or PAE 5–6 days (*Figure 3E*). Similarly, the expression levels of vitellogenin (Vg), a key downstream component of JH signaling triggering ovary development in locusts (*Song et al., 2014*), were prominently increased in fat body and ovary of females aged at PAE 3–4 days upon 4-VA stimuli (*Figure 3F–H*). By comparison, JH titer did not significantly change in Or35⁻/⁻ females exposed to 4-VA, contrast to over twofold increase in WT females (*Figure 4A*). Similar patterns were observed for the expression levels of Vg in fat body (*Figure 4B* and *Figure 4—figure supplement 1A*) and ovary (*Figure 4C*). To verify the critical roles of the JH/Vg pathway in mediating the effect of 4-VA, we further carried out rescue experiments by the injection of JH analog (methoprene) in Or35⁻/⁻ females. Methoprene-injected Or35⁻/⁻ females displayed more uniform sexual maturation (*Figure 4D*). Meanwhile, the expression levels of Vg in fat body and ovary significantly increased in methoprene-injected Or35⁻/⁻ females (*Figure 4E and F*, and *Figure 4—figure supplement 1B*). Results of rescue experiments on WT females indicated that inhibition of JH synthesis by destroying CA using precocene I blocked the 4-VA-accelarated female sexual maturation and Vg expression, which could be recovered by JH III application after precocene treatment (*Figure 4G–I*). These results provide clear evidence that the JH/Vg signaling pathway can mediate the time-dependent accelerating effects of 4-VA on sexual maturation synchrony in female locusts.

## Discussion

Our current study demonstrates conclusively that aggregation pheromone, 4-VA, acts to promote female maturation synchrony in locusts. The pheromone is abundantly released from gregarious male adults and speeds up oocyte development of females aged at PAE 3–4 days through activating JH synthesis and vitellogenesis (*Figure 5*). Our findings highlight a 'catch-up' strategy of reproductive synchrony by a time window effect combined with extra- and endo-signals in group-living animals.

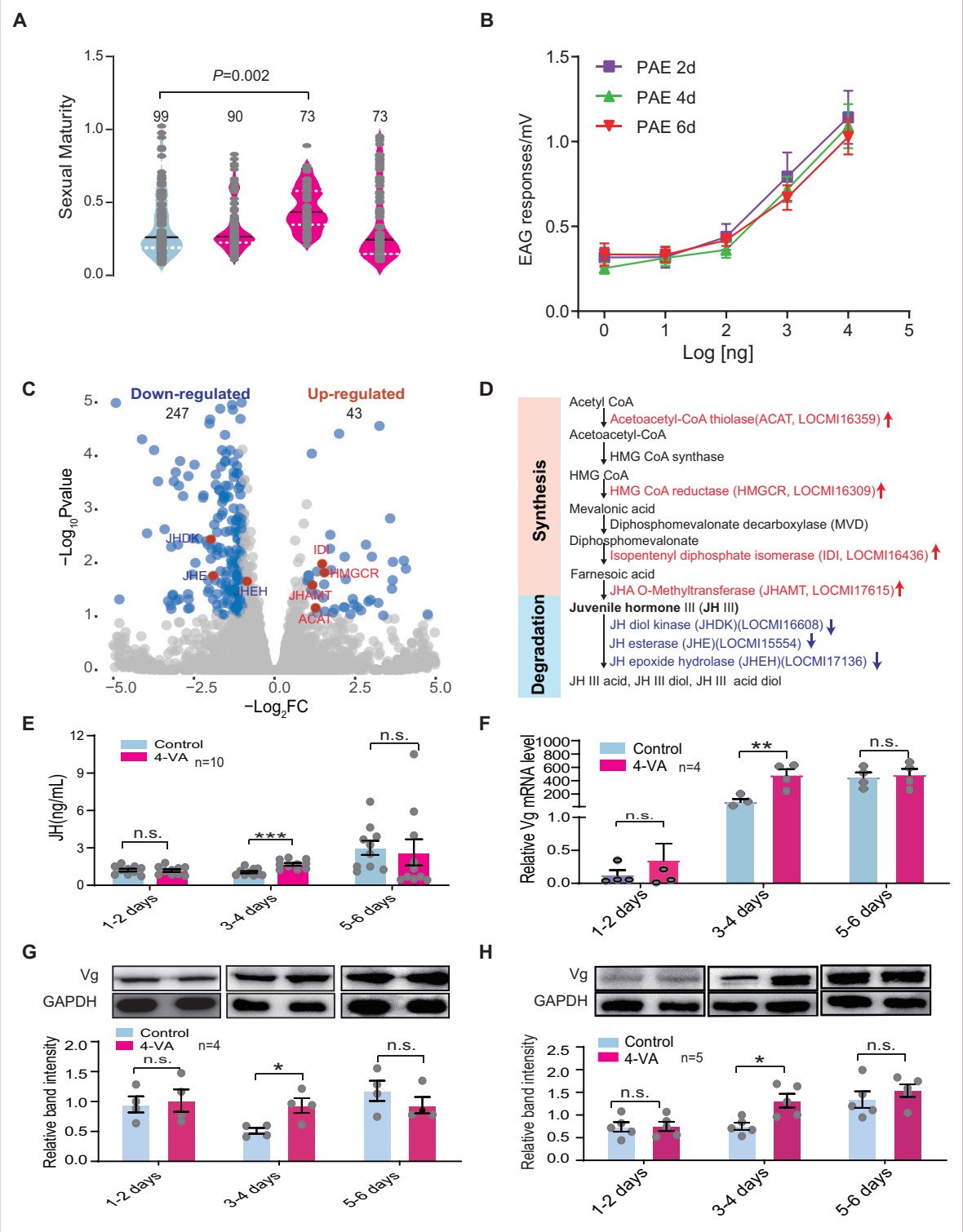

**Figure 3.** 4-Vinylanisole (4-VA) promotes sexual maturation synchrony in young females by enhancing juvenile hormone/vitellogenin (JH/Vg) signaling pathway at post adult eclosion (PAE) 3–4 days. (**A**) The 4-VA effects on sexual maturity of gregarious virgin females at different developmental stages. (**B**) Dosage effects on electroantennography (EAG) responses of females to 4-VA at different developmental stages. EAG responses to 4-VA with different concentrations were recorded in the antennae of female adults aged at PAE 2 days, PAE 4 days, and PAE 6 days, respectively (n = 7–11). (**C**) Volcano plot of RNA-seq in the corpus cardiacum-corpora allatum (CC-CA) complex of gregarious females after 4-VA stimulation at PAE 3–4 days. Red dots indicate genes related to JH metabolism. (**D**) Expression changes of JH metabolism-related genes in the CC-CA by 4-VA stimulation. Red and blue indicate upregulated and downregulated, respectively. (**E**) JH titers in the hemolymph, (**F**) the mRNA levels, (**G**) the protein levels of Vg in the fat

*Figure 3 continued on next page*

*Figure 3 continued*

body, and (**H**) the protein levels of Vg in the ovary of gregarious females after 4-VA stimulation at different developmental stages. Dark lines in violin plot indicate median value. White dotted lines indicate upper and lower quartiles, respectively; columns show means ± SEM. Consistency analysis of maturity was analyzed using Levene's test. The mRNA and protein levels were analyzed using Student's t-test. The number of biological replicates and p values were shown in the figures. n.s., not significant.

The online version of this article includes the following source data and figure supplement(s) for figure 3:

**Source data 1.** Raw data for sexual maturity, juvenile hormone (JH) titer, gene expression, and protein level in 4-vinylanisole (4-VA)-treated females.

**Figure supplement 1.** Effects of gregarious males with different ages on maturation synchrony of female aged at post adult eclosion (PAE) 3–4 days.

**Figure supplement 1—source data 1.** Raw data for sexual maturity of females reared with gregarious males with different ages.

**Figure supplement 2.** Enrichment of differentially expressed genes (DEGs) and Kyoto encyclopedia of genes and genomes (KEGG) in the fat body of gregarious females after 4-vinylanisole (4-VA) stimulation at different developmental stages.

**Figure supplement 3.** Peripheral electrophysiological responses of female locusts to 4-vinylanisole (4-VA).

**Figure supplement 3—source data 1.** Raw data for electrophysiological responses of female locusts to 4-vinylanisole (4-VA).

**Figure supplement 4.** The mRNA levels of *LmigOr35* during post adult eclosion (PAE) 1–8 days.

**Figure supplement 4—source data 1.** Raw data for mRNA levels of *LmigOr35* during post adult eclosion (PAE) 1–8 days.

**Figure supplement 5.** Volcano plot of RNA-seq data in the brain of females treated by 4-vinylanisole (4-VA) at post adult eclosion (PAE) 3–4 days.

**Figure supplement 6.** The mRNA level of JHAMT and JHE upon 4-vinylanisole (4-VA) treatment.

**Figure supplement 6—source data 1.** Raw data for mRNA level of JHAMT and JHE upon 4-vinylanisole (4-VA) treatment.

Reproduction synchrony involves consistence in maturation, mating, and egg laying, among which sexual maturation synchrony serves as the most foundational step for oviposition uniformity (*Hassanali et al., 2005*). Extremely high energy cost for female reproduction could restrict migration to pre-, post-, or inter-oviposition period in locusts, thus have crucial effects on collective movement of local populations (*Min et al., 2004*). Given this, a balance of sexual maturation timing among female members presents an essential subject for maintenance of locust swarms. We here demonstrated that young female adults reared with older gregarious male adults show faster and more synchronous sexual maturation in the migratory locust, supporting the accelerate role of crowding in sexual maturation of females (*Guo and Xia, 1964*; *Norris and Richards, 1964*). Together with the accelerating effects on immature male sexual maturation induced by older gregarious male adults reported previously (*Torto et al., 1994*; *Mahamat et al., 2011*), young adults of both sexes lived in gregarious conditions prefer more synchronous maturation than individuals reared in solitary. The consistent maturation in both sexes will greatly reduce intra- and inter-sexes competitions for mate selection and thus ensures reproductive synchronous in whole locust populations. We demonstrated that a single minor component (4-VA) of the volatiles abundantly released by gregarious male adults is sufficient to induce the maturation synchrony of female adults. By comparison, four volatiles (benzaldehyde, veratrole, PAN, and 4-vinylveratrole) showed stimulatory effects on male maturation (*Mahamat et al., 2011*). Thus, there might exist a sex-dependent action modes of maturation-accelerating pheromones: multi-component pheromones for males and single active component for females, possibly due to different selective pressures between two sexes in response to social interaction. Further exploration will be performed to confirm this hypothesis by determining whether 4-VA has maturation-accelerating effects on male adults in the migratory locust in future.

We prefer that 4-VA acts as a critical multi-functional pheromone for the formation of large locust swarms. Earlier, we have demonstrated that 4-VA is mainly released by gregarious nymphs and male adults (*Wei et al., 2017*) and can induce strong attraction behavior of both gregarious and solitarious phases (*Guo et al., 2020*), indicating its releaser pheromone role in in keeping locust individuals living together. Meanwhile, the present study shows that 4-VA, acting as a primer pheromone, promotes the maturation synchrony of young female adults, which might facilitate simultaneous oviposition and egg hatching to reduce the predation risk of an individual via the dilution effect (*Ward and Webster, 2016*). Thus, a dual role of 4-VA, including both primer and releaser pheromones, could be proposed in triggering the formation of locust swarms. The maintenance and coordination of locust swarming require elaborate communication mechanisms behind the interaction among individuals (*Pener and Simpson, 2009*; *Wang and Kang, 2014*). It is likely an effective and optimized strategy of group-living animals to use a single chemical pheromone to elicit both behavioral and endocrine responses

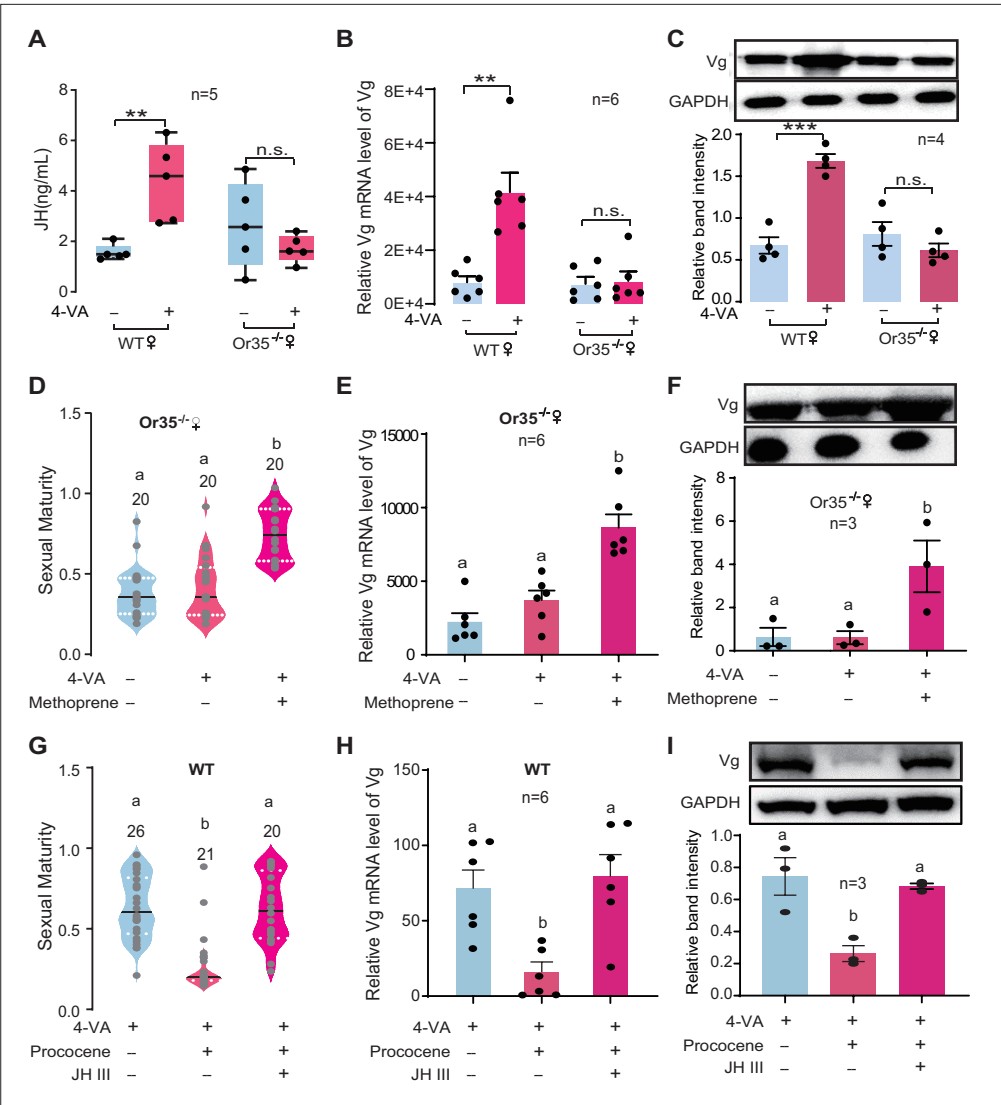

**Figure 4.** Juvenile hormone/vitellogenin (JH/Vg) pathway indeed mediates the stimulatory effects of 4-vinylanisole (4-VA) on female sexual maturation. (**A**) JH titers in the hemolymph of wild-type (WT) and Or35⁻/⁻ females after stimulation by 4-VA at post adult eclosion (PAE) 3–4 days. (**B**) The mRNA levels of *Vg* in the fat body and (**C**) protein levels of Vg in the ovaries of WT and Or35⁻/⁻ females after stimulation by 4-VA at PAE 3–4 days. (**D**) The effects of JH analog treatments on the maturity of Or35⁻/⁻ females exposed to 4-VA. (**E**) The mRNA level of *Vg* in the fat body and (**F**) protein level of Vg in the ovary of Or35⁻/⁻ females with 4-VA stimulation and JH analog treatments at PAE 3–4 days. (**G**) Sexual maturity in WT females treated by 4-VA, precocene I, and JH III. (**H**) The mRNA level of *Vg* in the fat body and (**I**) protein level of Vg in the ovary of WT females treated by 4-VA, precocene I, and JH III. All insects used were virgin females. Boxplots depict median and upper and lower quartiles. Lines in droplet diagram indicate median value; columns show means ± SEM. One-way ANOVA, p < 0.05. Columns labeled with different letters indicate a significant difference between these groups. The number of biological replicates is shown in the figure.

The online version of this article includes the following source data and figure supplement(s) for figure 4:

**Source data 1.** Raw data for juvenile hormone (JH) titer, gene expression, protein level, and sexual maturity in wild-type (WT) and Or35⁻/⁻ females.

**Figure supplement 1.** Validation the role of Or35 in 4-vinylanisole (4-VA)-enhanced vitellogenin (Vg) expression in the fat body of female locusts.

**Figure supplement 1—source data 1.** Raw data for vitellogenin (Vg) expression in the fat body of female locusts stimulated by 4-vinylanisole (4-VA).

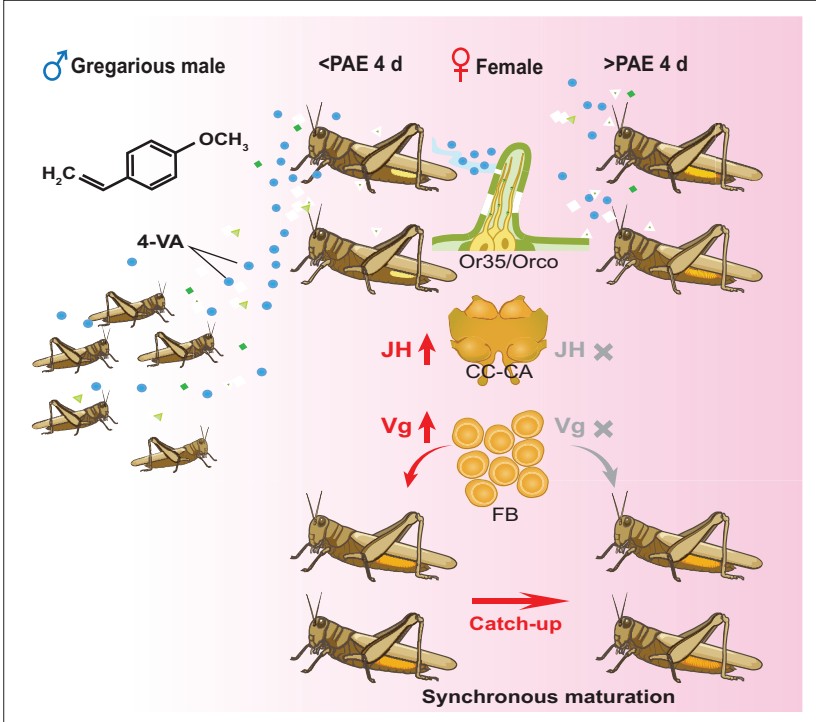

**Figure 5.** Schematic mechanisms underlying 4-vinylanisole (4-VA)-induced synchrony of female sexual maturation. 4-VA released from gregarious male locusts can significantly accelerate the ovary development of females with less-developed ovaries (approximately before post adult eclosion [PAE] 4 days) but not well-developed ovaries (after PAE 4 days). Mechanistically, after recognition by Or35 expressed in antennae, 4-VA promoted juvenile hormone (JH) synthesis in the CC and vitellogenesis in the fat body, thus accelerating female sexual maturation. The time-dependent stimulatory effects of 4-VA on ovary development finally led to the synchrony of female sexual maturation. CC-CA, corpora cardiaca and corpora allata; FB, fat body.

in conspecifics (*Rekwot et al., 2001*). The action of 4-VA displays a remarkable context-dependent manner, such as phase-, sex-, dose-, and time-dependent, reflecting physiological adaption of locusts to the highly dynamic nature of population density.

A dose-dependent manner was found for the maturation synchrony effect of 4-VA. We find that only gregarious males aged after PAE 3 days have the accelerating effects on female maturation synchrony, which may be attributed to their significantly increased 4-VA content during adult development. Although gregarious nymphs (the fifth instar) and female adults can release relatively small amount of 4-VA (*Wei et al., 2017*), they did not promote female maturation based on our current results. Thus, the accelerating effects on female maturation synchrony induced by gregarious male adults may depend largely on 4-VA content they released. The ineffectiveness of the fifth nymphs and females in maturation acceleration of female adults may due to their low 4-VA content under efficient threshold. In fact, the fifth nymphs have been shown to display inhibiting effects on male maturation in *S. gregaria* (*Assad et al., 1997*). Therefore, the mechanisms underlying pheromone-mediated sexual maturation may differ between different locust species. Recently, 4-VA has been identified in the volatiles released by male adults of *S. gregaria* (*Torto et al., 2021*), whether this volatile has maturation-accelerating effect in this locust species needs further validation.

Our results reveal that JH signaling pathway presents as the critical endocrinal factor mediating the accelerating effect of 4-VA on female maturation. This finding is consistent with the role of JH as the major gonadotropin modulating Vg biosynthesis in the fat body and its uptake by the growing oocytes in the migratory locust (*Jindra et al., 2013*; *Guo et al., 2014*; *Song et al., 2014*). It is also supported by the significance of CA (a major JH biosynthesis tissue) in pheromone-induced maturation process in the desert locust (*Odhiambo, 2009*). Interesting, it has been suggested that the release of the maturation-accelerating pheromone by adult males is under the control of CA (*Loher, 1997*). Thus, there should be a complex feedback interaction between 4-VA and JH signaling pathway. Extensive

studies have established the central roles of JH signaling in mediating the effects of social interactions on reproduction in different kinds of insect species, including eusocial insects (*Robinson and Vargo, 1997*; *Korb, 2015*), the burying beetle, *Nicrophorus vespilloides* (*Engel et al., 2016*), the German cockroach, *Blattella germanica* (*Uzsák and Schal, 2012*), and so on. Such an interaction between social clues and internal hormonal signals that coordinates ovary development is also common among group-living vertebrates (*Drickamer, 1977*; *McClintock, 1978*; *Berger, 1992*).

We demonstrate that 4-VA stimulates sexual maturation of young females within a distinct developmental time window. Compared to the females aged at PAE 1–2 days and 5–6 days, the females aged at PAE 3–4 days were more sensitive to 4-VA stimuli. This point was strongly supported by several lines of evidence from temporal-dependent comparisons of oocyte development, gene expression profiles, JH titer, as well as Vg biosynthesis. It has been shown that JH titers, Vg expression, the size of terminal oocytes, dramatically increased at PAE 3–4 days, implies the PAE 3–4 days is an essential time window for JH-regulated ovary development in female locusts (*Luo et al., 2017*; *Wu et al., 2018*). The finding that 4-VA accelerates maturation of less-developed females rather than more-developed females supports a 'catch-up' model in achievement of female maturation synchrony in locusts. We find that the release of 4-VA by gregarious males continuously increased after adult eclosion, with maximal 4-VA release at PAE 8 days. The age of maximal 4-VA production outwardly seems to be unmatched with the sensitive developmental stage to 4-VA of females (PAE 3–4 days). In insects, it is very common for males to mature earlier than females (*Alonzo, 2013*). In the locust, male adults also display earlier sexual maturation for several days, compared to females. In given locust population, individuals successively emerge to adults in a couple of days. Therefore, age-dependent increase in 4-VA release in gregarious male adults presents a persistent stimulus for less-developed young female adults, and thus maximizes maturation synchrony of female locusts, which could reduce male competitions for mate selection.

Peripheral and central neural sensitivity to olfactory clues have been demonstrated to vary with developmental stages or physiological statuses (*Guo et al., 2011*; *Gadenne et al., 2016*). Given this, sensory processing sensitivity or JH biosynthesis activity might be involved in the stage-dependent sensitivity of females to 4-VA stimuli. However, peripheral olfactory neuron might not be involved in the stage-specific sensitivity to 4-VA stimuli, because we did not detect significant changes of peripheral electrophysiological response during female ovary development. A possible explanation is that signaling factors responsible for JH synthesis might be turned on specifically at Mid-PAE of female locusts upon 4-VA stimuli, such as GPCRs and transcription factors (*Bendena et al., 2020*). Although there are only a few DEGs in the brain of females exposed to 4-VA, we cannot exclude the involvement of the central nerve system pathway by other regulatory mechanisms, for example, neurotransmitter release, or post-transcription regulation (*Nouzova et al., 2018*). Further studies should elucidate detailed mechanisms of the linking between 4-VA and JH biosynthesis in female locusts.

In summary, we revealed a catch-up strategy of female reproductive synchrony in locust swarms, whereby 4-VA acts as a maturation-accelerating pheromone hastening less-developed females through triggering JH biosynthesis. Our findings provide novel insight into the mechanisms underlying individual interaction during aggregation in group-living animals.

## Materials and methods
### Experimental insects
All insects used in experiments were reared in the same locust colonies at the Institute of Zoology, Chinese Academy of Sciences, Beijing, China. Briefly, gregarious locusts were reared in cages (30 cm × 30 cm × 30 cm) with 800–1000 first-instar insects per cage in a well-ventilated room. Solitarious locusts were individually raised in a ventilated cage (10 cm × 10 cm × 25 cm). All locusts were cultured under the following conditions: a L14:D10 photoperiod, temperature of 30°C ± 2°C, relative humidity of 60% ± 5%, and a diet of fresh greenhouse-grown seedlings and bran.

### Oocyte length measurement
All insects used for sexual maturity determination were virgin females. The ovary of tested females was dissected and placed in locust saline, and the terminal oocytes were isolated. The lengths of terminal oocytes were photographed and measured under the Leica DFC490 stereomicroscope

(Leica, Germany). Given that the maximum length of terminal oocytes in gregarious females is much longer than that in solitarious females (*Chen et al., 2015*), the sexual maturity was presented as the length of terminal oocyte relative to the maximum length.

## Recording of the distribution of first oviposition time in female locusts

Individuals between 0 and 24 hr after adult molting are referred as PAE 1 day adults, with each subsequent day representing an additional 24 hr period. Given that both gregarious and solitarious locusts begin to mate at PAE 6 days, the first oviposition time was recorded after PAE 6 days. For gregarious locusts, 10 females and 10 males at PAE 6 days were placed in a cage (30 cm × 30 cm × 30 cm). The females were individually marked, and their first oviposition times were recorded by collecting egg pods every 4 hr per day after mating. Females those laid new eggs could be easily distinguished by much thinner abdomen with white foam around ovipositor. For solitarious locusts, each female was reared together with a single male at PAE 6 days, and the first oviposition time was recorded by collecting egg pods every day after mating. To ensure the consistency of mating age in gregarious and solitarious locusts, females that did not successfully mate within 24 hr after paired rearing were excluded in both phases. The distribution curve of the first oviposition time was calculated based on data collected from all females.

## The effects of conspecifics interaction on female sexual maturation

Given that the difference of female sexual maturation synchrony between gregarious and solitarious phases appeared at PAE 6 days, the lengths of terminal oocytes of virgin females were detected at PAE 6 days after each treatment in subsequent experiments. For the stimulation of gregarious females, 10 gregarious females were reared with 10 gregarious males or 10 gregarious females in a same cage (15 cm × 15 cm × 10 cm) from PAE 1 to 6 days (*Figure 1—figure supplement 2A*). For the stimulation of solitarious females, one solitarious female was reared with one solitarious male or one solitarious female in a same cage (15 cm × 15 cm × 10 cm) from PAE 1 to 6 days (*Figure 1—figure supplement 2B*). The ovaries of treated females were dissected in locust saline and the lengths of terminal oocytes were measured as described above.

## The effect of locust volatiles on female sexual maturation

To determine the effect of locust volatiles on female sexual maturation, virgin female adults were separately reared with females or male adults by a breathable partition. For gregarious phase, 10 gregarious females were reared with 10 gregarious males or 10 gregarious females in a breathable partition cage (15 cm × 15 cm × 10 cm) from PAE 1 to 6 days (*Figure 1—figure supplement 2C*). For solitarious phase, one solitarious female was reared with 10 gregarious males or 10 solitarious males in a breathable partition cage (15 cm × 15 cm × 10 cm) from PAE 1 to 6 days (*Figure 1—figure supplement 2D*). The ovaries of treated females were dissected in locust saline and the lengths of terminal oocytes were measured as described above.

## SPME and GC-MS-MS

The volatiles of gregarious male adults, gregarious female adults, and solitarious male adults at PAE 1, 2, 4, 6, 8 days were collected by solid phase microextraction (SPME) for 30 min following our previously study (*Wei et al., 2017*). In detail, a fiber (PDMS/DVB 65 µm) was introduced into a glass jar (10.5 cm high ×8.5 cm internal diameter) to absorb odors. The SPME volatiles collected from an empty glass jar for 30 min served as the control. Eight biological replicates were performed for each treatment. The fibers with absorbed odors were subjected to chemical analyses with GC-MS/MS. A Bruker GC system (456-GC) coupled with a triple quadrupole (TQ) mass spectrometer (Scion TQ MS/MS, Inc, German) equipped with an DB-1MS column (30 m × 0.25 mm ID ×0.25 µm film thickness, Agilent Technologies) was used to quantify the volatile compounds in the SPME samples. Bruker chemical analysis MS workstation (MS Data Review, Data Process, version 8.0) was used to analyze and process the data. Mixed samples consisting of standard compounds in different dosages (0.1, 1, 5, 10, and 20 ng/µl) were used as external standards to develop the standard curves to quantify the volatiles. The same thermal program and Multiple Reaction Monitoring (MRM) method were used for standard compound detection.

## Odor treatment assay

For mixture treatment, 10 gregarious virgin females were stimulated by the mixed odor blend (the concentrations of PAN, guaiacol, 4-VA, vertrole, and anisole were 1, 10, 3, 2, 3 µg/µl, respectively) or paraffin oil from PAE 1 to 6 days. In detail, a breathable vial containing the mixture or paraffin oil was placed with 10 virgin females in a cage for 6 days. The vial was replaced by newly diluted compounds every day. The 4-VA treatment assay was performed by the same method, and the dose of 4-VA used is 100 ng/µl, and the concentration of 4-VA released was measured as 3.1–40 ng/0.5 hr within 24 hr exposure.

To determine the time-dependent effect of 4-VA, control females were treated by paraffin oil from PAE 1 to 6 days. In parallel, paraffin oil was placed by 4-VA at PAE 1–2, 3–4, 5–6 days, respectively. The ovaries were dissected and the lengths of terminal oocytes were measured as described above. The brains, CC-CA, and fat body of females were dissected and stored immediately in liquid nitrogen for further experiments.

## EAG assays

An aliquot of odor was dissolved in paraffin oil (w/v) and loaded with 10 µl on a 5 × 40 mm filter paper strip (Whatman), which was placed inside a Pasteur pipette. This odor was used on subsequent EAG assay. Hexane was tested as negative controls. The antennae of the adult locusts were cut at the bases of the flagella and distal antennal. Segments were cut off 2 mm and then fixed between two electrodes with electrode gel Spectra 360 (Parker, Orange, NJ). The EAG signals were amplified, monitored, and analyzed with the EAG-Pro software (IDAC4, Syntech, the Netherlands; EAG software v2.6c). A continuous air flow of 30 ml/s was produced by a stimulus controller (Syntech CS-05). Stimulation duration was 1 s and the intervals were 1 min. The blank was applied at the start and end of the stimulation series. The average EAG amplitude was subtracted from that of the blank.

## SSRs assay

SSRs were recorded and analyzed, and stimuli were prepared as previously described (*Li et al., 2016*). The locust was placed in a plastic tube 1 cm in diameter, and its head and antennae were fixed with dental wax. A tungsten wire electrode was electrolytically sharpened by 10% $NaNO_2$. The recording electrode was inserted into the bottom of the sensilla through a micromanipulator (Narishige, Japan). The reference electrode was inserted into the eye. Recording electrodes were connected to amplifiers (IDAC4, Syntech, the Netherlands). The frequency variation of each pulse at 0.2 s was calculated by using automatic frequency meter software. Signals were recorded for 10 s, starting 1 s before stimulation. The preparation is held in a humidified continuous air flow delivered by the Syntech Stimulus controller (CS-55 model, Syntech) at 1.4 l/min. Chemical substances as SSR stimulants included mineral oil as the blank, which was used to dilute the 4-VA at 1, 10, 100, 1000, 10,000, 100,000, 1,000,000 ng/µl, respectively. A piece of filter paper (Whatman, UK) was placed in a 15 cm Pasteur glass tube and 10 µl of volatile solution was added to the filter paper. Responses were calculated by counting the number of action potentials 1 s after stimulation.

## Total RNA extraction, RNA-seq, and quantitative real-time PCR

Total RNA from different tissues were extracted using the TRIzol reagent (Invitrogen TRIzol Reagent, Cat. 15596026) and treated with DNase I following the manufacturer's instructions.

For RNA-seq, three independent replicates were performed for each sample. The RNA-seq data reported here have been deposited in the Genome Sequence Archive (Genomics, Proteomics & Bioinformatics 2017) in National Genomics Data Center (Nucleic Acids Res 2020), Beijing Institute of Genomics (China National Center for Bioinformation), Chinese Academy of Sciences, under accession number CRA003038 that are publicly accessible at https://bigd.big.ac.cn/gsa.

## RNA integrity

cDNA libraries were prepared according to Illumina's protocols. Raw data were filtered, corrected, and mapped to locust genome sequence using TopHat2 software. The number of total reads was normalized by multiple normalization factors. Transcript levels were calculated using the reads per kb million mapped reads criteria. The differences between the test and control groups were represented by p values. DEGs were detected by using edgeR package with significance levels at $p < 0.05$. Principal

component analysis (PCA) was accomplished using the princomp and pca functions. Enrichment analysis of the Gene Ontology (GO) was carried out based on an algorithm presented by GOstat.

For qPCR, cDNA was reverse transcribed with 2 µg of total RNA using M-MLV Reverse Transcriptase (Promega, Madison, WI). The relative mRNA levels of targeting genes were quantified by Real Master Mix Kit (Tiangen) with LightCycler 480 instrument (Roche). Melting curve analysis was performed to confirm the specificity of amplification. The primers used for qPCR were presented in *Supplementary file 2*.

## Protein preparation and Western blot analysis

Ovaries and fat body of tested females were collected and homogenized in TRIzol reagent (5 individuals/sample, 6 biological repeats/treatment). Total protein was extracted following manufacturer's instructions. Total protein (100 µg) were separated by gel electrophoresis and then transferred onto polyvinylidene difluoride membranes (Millipore). Non-specific binding sites on the membranes were blocked with 5% bovine serum albumin. The blots were incubated with the primary antibodies (rabbit anti-Vg serum, 1:500, Beijing Protein Innovation Co., Ltd., BPI) in TBST overnight at 4°C. After incubation, the membranes were washed, incubated with anti-rabbit IgG secondary antibody (1:5000) (EASYBIO, China) for 1 hr at room temperature, and then washed again. Protein bands were detected by chemiluminescence (ECL kit, CoWin). The antibodies were stripped from the blots, re-blocked, and then probed with an anti-GAPDH antibody (1:5000) (*Wang et al., 2013*). Protein bands were detected by chemiluminescence (ECL kit, Thermo Scientific). The intensities of the Western blot signals were quantified using densitometry.

## JH titer measurement

Twenty microliters of hemolymph were added to a 1.5 ml tube with 100 µl of 70% methanol and thoroughly mixed. Then, 200 µl of hexane was added to the solution and thoroughly mixed again. The mixture was centrifuged at 5000 rcf for 10 min at 4°C. Then, 150 µl of supernatant was placed into a new tube, and the JH precipitate was dried by nitrogen. The JH precipitate was dissolved in 50% methanol, mixed by vortexing, and centrifuged at 13,000 rpm for 10 min at 4°C. JH III in the supernatant was detected using the rapid resolution liquid chromatography system (ACQUITY UPLC I-Class, Waters, Milford, MA).

An ACQUITY UPLC BEH C18 column (50 × 2.1 mm, 1.7 µm) was used for LC separation. The autosampler was set at 10°C, using gradient elution with 0.1% formic acid methanol as solvent A and 0.1% formic acid water as solvent B. The flow rate was set at 0.2 ml/min. Mass spectrometry detection was performed on an AB SCIEX Triple Quad 4500 (Applied Biosystems, Foster City, CA) with an electrospray ionization source (Turbo Ionspray). The detection was performed in positive electrospray ionization mode. The [M + H] of the analyte was selected as the precursor ion. The quantitation mode was MRM mode using the mass transitions (precursor ions/product ions). The MRM (m/z) of JH III was 267.2/235.2. Data acquisition and processing were performed using AB SCIEX Analyst 1.6 Software (Applied Biosystems).

## JH rescue experiment

For rescue experiment in Or35 mutants, the active JH analog, *S*-(+)-methoprene (Santa Cruz Biotech Dallas, TX) was topically applied to the pronotum of locusts (150 µg per locust) from PAE 3 to 4 days according to previously published work (*Song et al., 2013*; *Wu et al., 2016*), and acetone was used as the control. Meanwhile, the treated females were stimulated with 4-VA. The treated females were dissected at PAE 6 day, and the lengths of terminal oocytes were measured as previously described. The ovaries and fat bodies of females at PAE 4 days were dissected and stored immediately in liquid nitrogen for further Western blot experiments.

For rescue experiment in WT females, precocene I (Sigma) dissolved in acetone (100 µg/µl) was added to the dorsal neck membrane of locusts (500 µg/locust) within 12 hr after eclosion to inactive the corpora allata. JH III dissolved in acetone (20 µg/µl) was topically applied at 100 µg per locust aged at PAE 3 day to restore the JH activity. All females were treated by 4-VA at PAE 3–4 days. The treated females were dissected at PAE 6 day, and the lengths of terminal oocytes were measured, the mRNA level and protein level of Vg were detected to validate the effect of JH in 4-VA-accelerated sexual maturation and vitellogenesis.

## Statistical analyses

For the measurement of oviposition time and sexual maturation, individuals were randomly allocated into experimental group and control group, and no restricted randomization was applied.

The data that do not meet normal distribution was excluded for the analysis of sexual maturity, mRNA levels, protein levels, as well as JH titer measurement. The distribution of the first oviposition time and the consistency of sexual maturation (represented by the length of terminal oocytes) were analyzed using Levene's test according to previous studies (*Rohner et al., 2013*; *He et al., 2016*). The mean value of the first oviposition time between two groups was analyzed using Student's t-test. One-way ANOVA followed by Tukey's post hoc test was used for multi-group comparisons. All data were statistically analyzed using GraphPad Prism 5 software and SPSS 17 software. All experiments were performed with at least three independent biological replicates.

## Acknowledgements

This study was supported by the National Natural Science Foundation of China (Grant no. 31930012, 32088102, and 32070497) and grants from Chinese Academy of Sciences (nos. 152111KYSB20180036) and Youth Innovation Promotion Association CAS (no. 2021079).

## Additional information

### Funding

| Funder | Grant reference number | Author |
| --- | --- | --- |
| National Natural Science Foundation of China | 31930012 | Xianhui Wang |
| National Natural Science Foundation of China | 32070497 | Li Hou |
| Chinese Academy of Sciences | 152111KYSB20180036 | Le Kang |
| Youth Innovation Promotion Association of the Chinese Academy of Sciences | 2021079 | Li Hou |
| National Natural Science Foundation of China | 32088102 | Le Kang |

The funders had no role in study design, data collection and interpretation, or the decision to submit the work for publication.

### Author contributions

Dafeng Chen, Conceptualization, Data curation, Investigation, Methodology, Writing – original draft; Li Hou, Data curation, Funding acquisition, Investigation, Methodology, Writing – original draft; Jianing Wei, Investigation, Methodology; Siyuan Guo, Data curation, Methodology, Software; Weichan Cui, Data curation, Investigation, Methodology; Pengcheng Yang, Methodology, Software; Le Kang, Conceptualization, Project administration, Supervision, Writing – review and editing; Xianhui Wang, Conceptualization, Funding acquisition, Project administration, Supervision, Writing – review and editing

### Author ORCIDs

Li Hou ⓘ http://orcid.org/0000-0001-6727-7053
Le Kang ⓘ http://orcid.org/0000-0003-4262-2329
Xianhui Wang ⓘ http://orcid.org/0000-0002-8732-829X

### Decision letter and Author response

Decision letter https://doi.org/10.7554/eLife.74581.sa1
Author response https://doi.org/10.7554/eLife.74581.sa2

## Additional files

### Supplementary files
• Supplementary file 1. List of genes related to juvenile hormone (JH) synthesis and degradation in the corpus cardiacum-corpora allatum (CC-CA) of female adults exposure to 4-vinylanisole (4-VA) at post adult eclosion (PAE) 3–4 days.
• Supplementary file 2. Primers used in qPCR analysis.
• Transparent reporting form

### Data availability
All data generated or analysed during this study are included in the manuscript and supporting file; Source Data files have been provided for Figures 1, 2, 3, and 4.

The following dataset was generated:

| Author(s) | Year | Dataset title | Dataset URL | Database and Identifier |
|-----------|------|---------------|-------------|-------------------------|
| Yang P | 2020 | locust RNA-Seq with 4-vinylanisole treatment | https://ngdc.cncb.ac.cn/gsa/browse/CRA003038 | bigd, CRA003038 |

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
