## [Editor Report]

This study describes an important aspect of the devastating swarming behaviour of locusts – how gregarious female locusts might synchronise oviposition. It uncovers a role for olfactory signaling. The authors find that the aggregation pheromone 4-vinylanisole released by gregarious males is instrumental in synchronization of female sexual maturation. This study will be useful for the understanding of swarming behaviour in locusts, and it will also interest those who work on behaviour and its modulation.

---

## [Decision Letter]

**Decision letter after peer review:**

Thank you for submitting your article "Aggregation pheromone 4-vinylanisole promotes the synchrony of sexual maturation in female locusts" for consideration by *eLife*. Your article has been reviewed by 3 peer reviewers, including Sonia Sen as the Reviewing Editor and Reviewer #1, and the evaluation has been overseen by K VijayRaghavan as the Senior Editor. The following individual involved in review of your submission has agreed to reveal their identity: Amir Ayali (Reviewer #2).

While appreciating the general rigor and breadth of this study, we had a few concerns, which you can hopefully address in a revision. These are listed below:

Essential revisions:

1. Solitary female data: Since the focus of this study is the effect of 4VA on sexual maturation of gregarious females we recommend that the authors focus on this. We are concerned that the solitary female data presented in figures 1D and F pose a potentially problematic comparison.

2. Mating status: Have the authors considered that mating might play a role in synchrony in oviposition, since, in other insects, sex peptide is known to influence JH release from CC-CA post mating? Could the authors test this by using virgin females?

3. Age mismatch: Could the authors discuss the age-mismatch between the age of maximal 4VA production by G-males and the age of maximum effect in females?

4. Concentration of 4VA: Could the authors address what concentrations of 4VA are used with respect to physiological concentrations of 4VA produced? Which dose represents the levels released by 10 adult males at PAE 3-4 days?

5. Introduction and Discussion: Please revisit these sections to incorporate literature that is relevant to this study but has not been referred to here.

6. Choice of 4VA: We assume that part of the decision to chase 4VA was based on prior work. If this is true, could the authors emphasise this. The manner in which this choice is currently presented raises some issues. For example, would not differential analysis of volatiles between gregarious females and males be a better (or an additional) comparison? In general, could the authors make clear their rational and methodology and for the selection of the volatiles?

7. As the haemolymph JH titres in figure 3E are quite low, could the authors also include GAPDH quantification (for this and the subsequent figures), as internal controls, perhaps as a supplementary figure?

8. Allatectomy: We recommend that the authors perform this in the JH rescue experiment.

*Reviewer #1 (Recommendations for the authors):*

The data are clean and support the claims made, and the manuscript is well written. We have a few comments for the authors.

– The two experiments that support the idea of the critical window are the acceleration of sexual maturity at 3-4 DPE (not younger or older) in response to 4VA, and the RNAseq experiment from age matched fat bodies to show that most transcriptomic changes occur at 3-4 DPE (not younger or older) in response to 4VA. While these are convincing. We found it surprising that the effect of 4VA at this age appears distinctly attenuated compared to that presented in figure 2C, which is also assayed upon the presentation of 4VA alone. Is this true? Are the 4VA concentrations at a physiological range? Could the authors support this central claim by performing this experiment with appropriately aged gregarious males, instead of the volatile?

– In Figure 2B, are the 4VA concentrations used of physiological relevance? We're not sure if we're interpreting Figure 2-sup 1 correctly, but it suggests that the concentrations used in the experiments are far higher than what's emitted by males. Can the authors show the physiological quantification of 4-VA and clarify whether the 4-VA stimulations used in all the subsequent experiments are within physiological ranges?

– We assume that the CC-CA don't accelerate JH sysnthesis in response to 4VA at an earlier age as JH haemolymph titre is unaffected at that age upon 4VA exposure. This would suggest that the CC-CA of both gregarious (and solitary?) females of this age must be primed to increasing JH synthesis in response to 4VA. How could this be? The authors discuss that it might be GPCRs. Have they considered mating to have a role, since, in other insects, sex peptide is known to influence JH release from CC-CA?

– In general, can they comment on role of mating on sexual maturity? They say that both gregarious and solitary males "begin" to mate at PAE 6 days (line 341), is it possible that the dramatic effect seen post- PAE 4d is due to increased mating? This is further supported by the fact that the sexual maturity in Figure 1C (G-females co-housed with G-males) is much higher than Fig1E (G-females exposed to G-males volatiles). Is this difference also significant in WT females in 1H and 1I?

– We found it interesting that S-females reared G-male volatiles have comparable sexual maturity as compared to G-females with G-males (Figure 1F). Perhaps this supports that mating primes them for an altered 4VA response?

– Since the authors claim that the critical window allows a subset of the females' eggs to catch up with older ones so as to have synchronous oviposition it would be nice to have an oviposition read out of this phenomenon. Have the authors tested this? Perhaps by excluding this age group from an experiment such as that seen in C? One would expect no catch up here and therefore a widely distributed oviposition.

– As the haemolymph JH titres in figure 3E are quite low, could the authors also include GAPDH quantification (for this and the subsequent figures), as internal controls, perhaps as a supplementary figure?

– Females of all ages respond similarly to 4VA. Despite that, only the female of a certain age accelerates oogenesis. This is very interesting! Do the authors want to consider presenting this as one of the main figures?

*Reviewer #2 (Recommendations for the authors):*

The abstract can be improved to better present the findings and their significance.

LL92-93: Anstey et al. has little to do with coordinated behavior. There are other much more relevant studies (e.g. Buhl, Ayali, others).

The introduction should do a much better job in presenting the advantages of reproductive synchrony for locusts.

LL 117-118 I would rather the authors would have first investigated whether there is any synchronization effect associated with the gregarious phase per se, only than adding the comparative information on the solitary phase (where no synchronization is to be expected).

The experimental procedures behind Figure 1 are not presented in a clear enough manner. For example, what exactly is the Sexual maturity index? Why not show length of the terminal oocyte?

LL 355-357 "To avoid the effects caused by asynchronous mating." Not clear.

LL 130-131 "by the removal of male adults in gregarious phase (Figure 1C), but not in solitarious phase (Figure 1D)." Not clear. What was the manipulation conducted on Solitary females how and why is it comparable to that conducted with the gregarious?

Figure 1C vs E: what explains the major difference in the response of the females?

Including the solitary locusts data in Figure 1 adds very little!

LL 148-150. I was under the impression that this was already done in previous studies

LL 211-212 In locusts there really is no CA-CC complex, like in other insects. The CA are easily distinguished and are those attributed with a role in JH/Vg signaling pathway. Not sure why were the CC included.

Figure 3F-H n=4?

Figure 5 – It is not clear what is the difference between the females in the bottom left vs. right.

LL 268-269 ?

*Reviewer #3 (Recommendations for the authors):*

I think the discussion lacks depth in relation to the biology of gregarious locusts because of the scope of the results which focused on only one locust sex (females). It would be more interesting to investigate sexual maturation in both sexes and the underlying mechanisms. One more thing, I think the authors may have missed the new literature on the composition of odors of the desert locust which reports 4-vinyl anisole as an adult-male specific volatile (see. https://doi.org/10.1016/j.jinsphys.2021.104296). Hence the statement by the authors, "Given that 4-VA has not been detected in S. gregaria (Torto et al., 1996), whether this volatile has maturation-accelerating effect in this locust species needs further validation," must be rephrased.

---

## [Author Response]

Essential revisions:1. Solitary female data: Since the focus of this study is the effect of 4VA on sexual maturation of gregarious females we recommend that the authors focus on this. We are concerned that the solitary female data presented in figures 1D and F pose a potentially problematic comparison.

We understand the reviewer’s concern. However, we think the solitarious phase data is very important for our findings. The aim of this study is to explore the mechanism underlying sexual maturation synchrony by comparing phase- and sex-dependent conspecific interactions in locusts. Given that solitarious males have no stimulatory effect on sexual maturation, phase-dependent comparison of volatile contents is helpful for us to screen candidate volatiles responsible for the acceleration of sexual maturation synchrony in females. According to the suggestion of Reviewer 2, we have added the comparison of volatile contents between gregarious males and gregarious females for effectively screening of targeting volatiles (see revised Figure 2A).

2. Mating status: Have the authors considered that mating might play a role in synchrony in oviposition, since, in other insects, sex peptide is known to influence JH release from CC-CA post mating? Could the authors test this by using virgin females?

We understand the reviewer’s concerns. We have indeed taken the effects of mating on sexual maturation and oviposition into consideration in current study. To eliminate the effects of different mating age on oviposition, only females successfully mate at 6-7 days post eclosion (PAE 6-7 days) were used in both gregarious and solitarious phases for measurement of synchrony in first oviposition time. In addition, only virgin females were used for in all the experiments related to measurement of sexual maturation synchrony in both phases. We have provided these method details in the Figure legend and Materials and methods (lines 372).

3. Age mismatch: Could the authors discuss the age-mismatch between the age of maximal 4VA production by G-males and the age of maximum effect in females?

We accept the reviewer’s suggestions. According to the suggestion, we have provided additional discussion on the age-mismatch between the age of maximal 4-VA production by G-males and the age of maximum effect in females. Details were shown as: “We find that the release of 4-VA by gregarious males continuously increased after adult eclosion, with maximal 4-VA release at PAE 8 days. The age of maximal 4-VA production outwardly seems to be unmatched with the sensitive developmental stage to 4-VA of females (PAE 3-4 days). In insects, it is very common for males to mature earlier than females (Alonzo, 2013). In the locust, male adults also display earlier sexual maturation for several days, compared to females. In given locust population, individuals successively emerge to adults in a couple of days. Therefore, age-dependent increase in 4-VA release in gregarious male adults presents a persistent stimulus for less-developed young female adults, and thus maximizes synchronous maturation of female locusts, which could reduce male competitions for mate selection” (lines 336-345).

4. Concentration of 4VA: Could the authors address what concentrations of 4VA are used with respect to physiological concentrations of 4VA produced? Which dose represents the levels released by 10 adult males at PAE 3-4 days?

Thanks. Our analysis revealed that the physiological concentration of 4-VA is 0.18-1.00 ng/0.5 h/locust in gregarious male adults aged at 4 days (1.80-10.00 ng/0.5 h for 10 individuals). Results from dosage effects of 4-VA on sexual maturity in females indicated that the effective dose is 100 ng/ul (the concentration of 4-VA released was 3.1−40 ng/0.5 h within 24 h exposure). Thus, the 4-VA volatile quantity used is within the physiological range of 4-VA production. Detailed information has been added to the methods part (line 431-432).

5. Introduction and Discussion: Please revisit these sections to incorporate literature that is relevant to this study but has not been referred to here.

We have revised the contents and replaced the literature following the reviewer’s suggestions. The importance of reproductive synchrony and the research progress in regulatory mechanisms underlying sexual maturation in locusts have been added to the introduction, details were shown as: “Locusts often form large swarms with hundreds to thousands of individuals, regarded as one of the most extraordinary examples of coordinated behavior (Ariel and Ayali, 2015, Buhl and Rogers, 2016). Depending on population density, locusts display striking phenotypic plasticity, with a cryptic solitarious phase and an active gregarious phase (Wang and Kang, 2014). Gregarious locusts, compared to solitarious conspecifics, show much higher synchrony in physiological and behavioral events, such as egg hatching and sexual maturation, as well as synchronous feeding and marching behaviors (Norris, 1954, Uvarov, 1977). Reproductive synchrony in gregarious locusts provides benefits for individuals in several aspects, such as more favorable microenvironment, lower risk of predation, efficiently forging, as well we more encounters with mates, therefore ensures high density conditions for the next generation, and is essential for maintenance of locust swarm (Beekman et al., 2008, Maeno et al., 2021). Some sort of vibratory stimulus, maternal microRNAs, and SNARE protein play important roles in the egg-hatching synchrony of gregarious locusts (Chen et al., 2015b, He et al., 2016, Nishide and Tanaka, 2016). It has been revealed that the presence of mature male adults has effectively accelerating effects on synchrony of sexual maturation of immature male and female conspecifics in two locust species, *Schistocerca gregaria* and *Locusta migratoria* (Norris, 1952, Loher, 1961, Guo and Xia, 1964, Norris, 1964). The accelerating effects of several prominent volatiles released by gregarious mature males in male maturation have been exampled in the desert locust. Four volatile pheromones (benzaldehyde, veratrole, phenylacetonitrile, and 4-vinylveratrole) have significantly stimulatory effects on sexual maturation of male adults, with phenylacetonitrile having the most pronounced effect. (Mahamat et al., 1993, Assad et al., 1997). However, how conspecific interaction affects female sexual maturation remains unclear and the pheromones those contribute to maturation synchrony of females have not been determined so far” (lines 92-114). Moreover, the importance of reproductive synchronization in female locusts as well as age mis-match between maximal 4-VA release from gregarious males and sensitive developmental stage to 4-VA of females have also been added in the discussion (see details in following section).

6. Choice of 4VA: We assume that part of the decision to chase 4VA was based on prior work. If this is true, could the authors emphasise this. The manner in which this choice is currently presented raises some issues. For example, would not differential analysis of volatiles between gregarious females and males be a better (or an additional) comparison? In general, could the authors make clear their rational and methodology and for the selection of the volatiles?

We understand the reviewer’s query. Actually, here the functional study of 4-VA in sexual maturation is not to chase our previous work (Guo et al., 2020, Nature, 4-vinylanisole is an aggregation pheromone in locusts). Instead, we aim to explore the mechanisms underlying sexual maturation synchrony by comparing phase- and sex-dependent conspecific interactions in locusts. Given that the volatiles released by gregarious males, rather than gregarious females and solitarious males, have the accelerate effects on female sexual maturation, a comparative analysis of volatile contents among these three groups (G-males, G-females, and S-males) was performed in the revision process (revised Figure 2A). Compared to volatiles released by G-females, and S-males, only five kinds of volatiles display significantly higher emission in G-males (PAN, guaicol, 4-VA, vertrole, and anisole). The roles of these five candidate volatiles in female sexual maturation were individually validated by removing the volatile from the stimulation blend one by one. The results showed that only the omission of 4-VA from the blends lost the accelerating effects on sexual maturation synchrony of gregarious females (Figure 2B). Based on these findings, we inferred that 4-VA abundantly released by gregarious male adults played major roles in promoting female sexual maturation synchrony. The logic of selection of candidate volatiles accelerating sexual maturation synchrony has been described as: “To identify key active compounds that can promote sexual maturation synchrony of female locusts, we compared the volatile emission dynamics of gregarious male adults, gregarious female adults, and solitarious male adults from PAE 1-8 days. In total, 14 chemicals were identified in the volatiles released by male adults (Figure 2A). Only 5 compounds displayed significantly higher abundance in gregarious-male adults, compared to volatile released by gregarious female adults and solitarious male adults, which showed no accelerating effects on female sexual maturation. Thus, these five kinds of gregarious male adult-abundant volatiles, including phenylacetonitrile (PAN), guaicol, 4-vinylanisole (4-VA), vertrole, and anisole (Figure 2A and Figure 2—figure supplement 1), might serve as candidate pheromones for female maturation acceleration” (lines 156-165).

7. As the haemolymph JH titres in figure 3E are quite low, could the authors also include GAPDH quantification (for this and the subsequent figures), as internal controls, perhaps as a supplementary figure?

The JH titers measured in this work is at the same level with that reported in previous work in locusts (0-5 ng/ml, Guo et al., 2020, PLoS Genet; Guo et al., 2019, FASEB Journal). And, GAPDH is commonly used as internal reference in western blot experiments examining the abundance of target proteins, not used for HPLC methods that we used for juvenile hormone quantification.

8. Allatectomy: We recommend that the authors perform this in the JH rescue experiment.

Thanks. Allatectomy is a classical method to abolish the capacity of CA for JH synthesis in many early studies. Besides, precocene treatment is now commonly used for effective inhibition of JH synthesis (Wu et al., 2016, The Journal of Biological Chemistry). During the revision process, we performed addition rescue experiments in WT females by inhibiting JH synthesis using Precocene (PI) before JH treatment. The results showed that PI treatment significantly inhibited sexual maturation rate and Vg expression in 4-VA-exposed WT females, whereas JH treatment post application can obviously recover the sexual maturation rate and Vg expression (Figure 4G-I).

Reviewer #1 (Recommendations for the authors):The data are clean and support the claims made, and the manuscript is well written. We have a few comments for the authors.– The two experiments that support the idea of the critical window are the acceleration of sexual maturity at 3-4 DPE (not younger or older) in response to 4VA, and the RNAseq experiment from age matched fat bodies to show that most transcriptomic changes occur at 3-4 DPE (not younger or older) in response to 4VA. While these are convincing. We found it surprising that the effect of 4VA at this age appears distinctly attenuated compared to that presented in figure 2C, which is also assayed upon the presentation of 4VA alone. Is this true? Are the 4VA concentrations at a physiological range? Could the authors support this central claim by performing this experiment with appropriately aged gregarious males, instead of the volatile?

Thanks for the reviewer’s comments. We understand the reviewer’s concerns about the seemingly nonconsistency of the effects of 4-VA treatments on sexual maturity between Figures including Figure 2C and Figure 3A. However, the comparison with the absolute value of sexual maturity between different figures is not meaningful because the data of different figures were obtained from independent experiments, in which their oocyte developments might vary due to batch effects. In each experiment, the control was designed for avoiding the batch effect. The significant effect of 4-VA treatment was demonstrated from the comparison with the control in all of experiments. So, it is not true that the effect of 4-VA at this age appears distinctly attenuated in Figure 3A compared to that presented in figure 2C, just due to batch effect. And the 4VA concentrations that we used should be at a physiological range. Based on our data form GC-MS-MS, the physiological concentration of 4-VA from ten gregarious male adults aged at 4 days is 1.8-10 ng/0.5 h. The concentration of 4-VA (100 ng/μl) released was measured as 3.1−40 ng/0.5 h within 24 h exposure. Thus, the 4-VA volatile quantity used is within the physiological range of 4-VA production. Detailed information has been added to the methods part (lines 431-432). To further clarify reviewer’s concerns, we have performed the experiment suggested by the reviewer by comparing the effects of gregarious males with different ages on female maturation. As shown in Figure 3—figure supplement 1, the maturation synchrony of females was significantly enhanced by gregarious males aged at PAE 3-4 days and PAE 5-6 days.

– In Figure 2B, are the 4VA concentrations used of physiological relevance? We're not sure if we're interpreting Figure 2-sup 1 correctly, but it suggests that the concentrations used in the experiments are far higher than what's emitted by males. Can the authors show the physiological quantification of 4-VA and clarify whether the 4-VA stimulations used in all the subsequent experiments are within physiological ranges?

The 4-VA volatile quantity that we used should be within the physiological range. The detailed reason can be seen above.

– We assume that the CC-CA don't accelerate JH synthesis in response to 4VA at an earlier age as JH haemolymph titre is unaffected at that age upon 4VA exposure. This would suggest that the CC-CA of both gregarious (and solitary?) females of this age must be primed to increasing JH synthesis in response to 4VA. How could this be? The authors discuss that it might be GPCRs. Have they considered mating to have a role, since, in other insects, sex peptide is known to influence JH release from CC-CA?

Thanks for the reviewer’s nice suggestions. We don’t think that mating plays a role in the accelerating effect of JH synthesis observed in our studies because only virgin females were used in all our experiments refers to sexual maturation examination, JH titer determination, and gene expression analysis (See the method for oocyte length measurement, lines 372-377). Certainly, it will be a valuable subject to explore how mating or sex peptides affect female sexual maturation and JH synthesis in future work.

– In general, can they comment on role of mating on sexual maturity? They say that both gregarious and solitary males "begin" to mate at PAE 6 days (line 341), is it possible that the dramatic effect seen post- PAE 4d is due to increased mating? This is further supported by the fact that the sexual maturity in Figure 1C (G-females co-housed with G-males) is much higher than Fig1E (G-females exposed to G-males volatiles). Is this difference also significant in WT females in 1H and 1I?

Thanks for the reviewer’s comments. Just seen our response above, in fact, virgin females were used in all experiments refers to sexual maturation examination in Figure 1. Therefore, the role of mating in phase-dependent sexual maturation in the current study could be excluded. And, about the variance of sexual maturity between figures, batch effects can be explained (Seen our above responses). To further validate the role of gregarious male adults rather than gregarious female adults in accelerating sexual maturation of females, we have repeated the experiments in Figure 1C. The results conform the maturation-accelerating effect of gregarious male adults (revised Figure 1C).

– We found it interesting that S-females reared G-male volatiles have comparable sexual maturity as compared to G-females with G-males (Figure 1F). Perhaps this supports that mating primes them for an altered 4VA response?

Thanks for the reviewer’s suggestion. Just seen our above response about the role of mating. In fact, we used virgin females in all experiments refers to sexual maturity examination to eliminate the potential effects of mating. To avoid the misunderstanding, we have added the information about mating status of females used in each experiment in the figure legends and methods.

– Since the authors claim that the critical window allows a subset of the females' eggs to catch up with older ones so as to have synchronous oviposition it would be nice to have an oviposition read out of this phenomenon. Have the authors tested this? Perhaps by excluding this age group from an experiment such as that seen in C? One would expect no catch up here and therefore a widely distributed oviposition.

We thank the reviewer’s suggestions. Female ovarian maturation is an essential prerequisite for oviposition, and is thus commonly used as an indicator for reproductive activity in locusts (Wu et al., 2016, J Bio Chem, Song et al., 2018, Development; Gijbels et al., 2019, Sci Rep). The prime objective of current study is to explore the regulatory mechanism underlying sexual maturation synchrony in locusts. The time window-dependent accelerating effects of 4-VA on sexual maturation of female locusts were conceivably supported by several lines of evidence, including significantly changed sexual maturity, gene expression profiles, JH synthesis, and vitellogenesis resulted from 4-VA treatment at PAE 3-4 days. Based on these findings, the further validation of time-window effects of 4-VA on oviposition might not be necessary in the current study.

– As the haemolymph JH titres in figure 3E are quite low, could the authors also include GAPDH quantification (for this and the subsequent figures), as internal controls, perhaps as a supplementary figure?

Thanks for the reviewer’s suggestion. In fact, the JH titers measured in this work is at the same level with that reported in previous work in locusts (0-5 ng/ml, Guo et al., 2020, PLoS Genetics; Guo et al., 2019,FASEB Journal ). And GAPDH is commonly used as internal reference in western blot experiments examining the abundance of target proteins. Because HPLC method was used for JH hormone quantification in this study, GAPDH is not useful for hormone quantification.

– Females of all ages respond similarly to 4VA. Despite that, only the female of a certain age accelerate oogenesis. This is very interesting! Do the authors want to consider presenting this as one of the main figures?

We appreciate the reviewer’s helpful suggestion. Now we have presented the EAG response to 4-VA at different development stage in the main figure (Figure 3B).

Reviewer #2 (Recommendations for the authors):The abstract can be improved to better present the findings and their significance.

The abstract has been revised as: “Reproductive synchrony generally occurs in various group-living animals. However, the underlying mechanisms remain largely unexplored. The migratory locust, *Locusta migratoria*, a worldwide agricultural pest species, displays synchronous maturation and oviposition when forms huge swarm. The reproductive synchrony among group members is critical for the maintenance of locust swarms and population density of next generation. Here, we showed that gregarious female locusts displayed more synchronous sexual maturation and oviposition than solitarious females and olfactory deficiency mutants. Only the presence of gregarious male adults can stimulate sexual maturation synchrony of female adults. Of the volatiles emitted abundantly by gregarious male adults, the aggregation pheromone, 4-vinylanisole, was identified to play key role in inducing female sexual maturation synchrony. This maturation-accelerating effect of 4-vinylanisole disappeared in the females of Or35^-/-^ lines, the mutants of 4-vinylanisole receptor. Interestingly, 4-vinylanisole displayed a time-window action by which mainly accelerates oocyte maturation of young females aged at middle developmental stages (3-4 days post adult eclosion). We further revealed that juvenile hormone/vitellogenin pathway mediated female sexual maturation triggered by 4-vinylanisole. Our results highlight a “catch-up” strategy by which gregarious females synchronize their oocyte maturation and oviposition by time-dependent endocrinal response to 4-vinylanisole, and provide insight into reproductive synchrony induced by olfactory signal released by heterosexual conspecifics in a given group”.

LL92-93: Anstey et al. has little to do with coordinated behavior. There are other much more relevant studies (e.g Buhl, Ayali, others).

We have added the references as the reviewer’s suggestion. Details were shown as “Locusts often form large swarms with hundreds to thousands of individuals, regarded as one of the most extraordinary examples of coordinated behavior (Ariel and Ayali, 2015, Buhl and Rogers, 2016)” (lines 93-94).

The introduction should do a much better job in presenting the advantages of reproductive synchrony for locusts.

We have revised the introduction as the reviewer’s suggestion: “Reproductive synchrony in gregarious locusts provides benefits for individuals in several aspects, such as more favorable microenvironment, lower risk of predation, efficiently forging, as well we more encounters with mates, therefore ensure high density conditions for the next generation, and is essential for maintenance of locust swarm (Beekman et al., 2008, Maeno et al., 2021). Some sort of vibratory stimulus, maternal microRNAs, and SNARE protein play important roles in the egg-hatching synchrony of gregarious locusts (Chen et al., 2015, He et al., 2016, Nishide and Tanaka, 2016). It has been revealed that the presence of mature male adults has effectively accelerating effects on synchrony of sexual maturation of immature male and female conspecifics in two locust species, *Schistocerca gregaria* and *Locusta migratoria* (Norris, 1952, Loher, 1961, Guo and Xia, 1964, Norris, 1964). The accelerating effects of several prominent volatiles released by gregarious mature males in male maturation have been exampled in the desert locust. Four volatile pheromones (benzaldehyde, veratrole, phenylacetonitrile, and 4-vinylveratrole) have significantly stimulatory effects on sexual maturation of male adults, with phenylacetonitrile having the most pronounced effect. (Mahamat et al., 1993, Assad et al., 1997). However, how conspecific interaction affects female sexual maturation remains unclear and the pheromones that contribute to maturation synchrony of females have not been determined so far” (lines 98-114).

LL 117-118 I would rather the authors would have first investigated whether there is any synchronization effect associated with the gregarious phase per se, only than adding the comparative information on the solitary phase (where no synchronization is to be expected).

As descripted above, the aim of this study is to explore the mechanism underlying sexual maturation synchrony by comparing phase- and sex-dependent conspecific interactions in locusts. The reproductive synchrony in gregarious might be not highlighted without comparison with solitarious locusts, including both first oviposition time and sexual maturation, although the mechanism studies were mostly performed in gregarious locusts in current work. Moreover, phase-dependent comparison of volatile contents is helpful for us to screen candidate volatiles responsible for the acceleration of sexual maturation synchrony in females.

The experimental procedures behind Figure 1 are not presented in a clear enough manner. For example, what exactly is the Sexual maturity index? Why not show length of the terminal oocyte?

Thanks for the reviewer’s comments. The sexual maturity index is expressed as the length of terminal oocyte relative to the maximum length. Detailed experimental procedures have been added in the revised method: “Given that the maximum length of terminal oocyte in gregarious females is much longer than that in solitarious females (Chen et al., 2015), the sexual maturity was presented as the length of terminal oocyte relative to the maximum length” (lines 375-377).

LL 355-357 "To avoid the effects caused by asynchronous mating." Not clear.

Thanks, we have revised as “To ensure the consistency of mating time in gregarious and solitarious locusts, females that did not successfully mate within 24 h after paired rearing were excluded in both phases” (lines 387-389).

LL 130-131 "by the removal of male adults in gregarious phase (Figure 1C), but not in solitarious phase (Figure 1D)." Not clear. What was the manipulation conducted on Solitary females how and why is it comparable to that conducted with the gregarious?

Thanks, we have revised as “We found that the maturation synchrony of terminal oocytes of females was significantly retarded by the removal of male adults in gregarious phase (Figure 1C). However, whether raised with either solitarious female or male does not affect sexual maturation of solitarious females (Figure 1D)” (137-140).

Figure 1C vs E : what explains the major difference in the response of the females? Including the solitary locusts data in Figure 1 adds very little!

Figure 1C showed that reared with gregarious males, not females, could significantly promotes sexual maturation synchrony of gregarious females; Figure 1E showed that stimulated by volatile emissions of gregarious males enhanced sexual maturation synchrony of gregarious females. For the solitary locust data in Figure 1 D and E, we provide further evidence indicating that the effective compounds for sexual maturation synchrony should be from gregarious male adults, but not solitarious males, or solitarious females. Based on this evidence, the phase- and sex-dependent comparisons of volatile contents were then used for screening candidate sexual maturation accelerating pheromone (Figure 2A).

LL 148-150. I was under the impression that this was already done in previous studies

In the previous work of our lab, Wei et al. analyzed the emission dynamics of locust volatiles associated with development, sexes, and phase changes (Wei et al., 2016, Insect science). However. the objective of this study is different from our previous work. In the current study, emission dynamics of locust adults during sexual maturation period (PAE 1, PAE 2, PAE4, PAE6, PAE8) were continuously monitored to identify candidate volatiles with maturation-accelerating effects. Besides, the age of experimental insects used in these two studies are different.

LL 211-212 In locusts there really is no CA-CC complex, like in other insects. The CA are easily distinguished and are those attributed with a role in JH/Vg signaling pathway. Not sure why were the CC included.

We accept the reviewer’s query. The aim of RNA-seq experiments is to investigate the effects of 4-VA on neuroendocrinal tissues involved in reproduction control, including brain, CC, and CA, not just CA. So, we did not dissect CC and CA tissues, separately. The data from RNA-seq demonstrated that the expression levels of genes related to JH-metabolism were significantly affected upon 4-VA treatment.

Figure 3F-H n=4?

Thanks, we have revised it. For Figure 3F and G, n=4; For Figure 3H, n=5. The numbers of biological replicate are indicated in the figures.

Figure 5 – It is not clear what is the difference between the females in the bottom left vs. right.

Thanks, we have revised the figure to make it clear to be understood. The females in the bottom left have similar sexual states to that of the females in the right, indicating more synchronous sexual maturation states among individuals.

LL 268-269 ?

We have removed the incomplete description.

Reviewer #3 (Recommendations for the authors):I think the discussion lacks depth in relation to the biology of gregarious locusts because of the scope of the results which focused on only one locust sex (females). It would be more interesting to investigate sexual maturation in both sexes and the underlying mechanisms. One more thing, I think the authors may have missed the new literature on the composition of odors of the desert locust which reports 4-vinyl anisole as an adult-male specific volatile (see. https://doi.org/10.1016/j.jinsphys.2021.104296). Hence the statement by the authors, "Given that 4-VA has not been detected in S. gregaria (Torto et al., 1996), whether this volatile has maturation-accelerating effect in this locust species needs further validation," must be rephrased.

Thanks for the reviewer’s suggestion. (1) we have provided additional discussion on sexual maturation synchrony in both sexes and the underlying mechanism. Details were shown as: “Reproduction synchrony involves consistence in maturation, mating, and egg laying, among which sexual maturation synchrony serve as the most foundational step for oviposition uniformity (Hassanali et al., 2005). Extremely high energy cost for female reproduction could restrict migration to pre, post, or inter oviposition period in locusts, thus have crucial effects on collective movement of local populations (Min et al., 2004). Given this, a balance of sexual maturation timing among female members presents an essential subject for maintenance of locust swarms. We here demonstrated that young female adults reared with older gregarious male adults show faster and more synchronous sexual maturation in the migratory locust, supporting the accelerate role of crowing in sexual maturation of females (Guo and Xia, 1964, Norris and Richards, 1964,). Together with the accelerating effects on immature male sexual maturation induced by older gregarious male adults reported previously (Torto et al., 1994, Mahamat et al., 2000), young adults of both sexes lived in gregarious conditions prefers more synchronous maturation than individuals reared in solitary. The consistent maturation in both sexes will greatly reduce intra- and inter-sexes competitions for mate selection thus ensures reproductive synchronous in whole locust populations. We demonstrated that a single minor component (4-VA) of the volatiles abundantly released by gregarious male adults is sufficient to induce the maturation synchrony of female adults. By comparison, four volatiles (benzaldehyde, veratrole, phenylacetonitrile, and 4-vinylveratrole) showed stimulatory effects on male maturation (Mahamat et al., 2000). Thus, there might exist a sex-dependent action modes on maturation-accelerating pheromones: multicomponent pheromones for males and single active component for females, possibly due to different selective pressures between two sexes in response to social interaction. Further exploration will be performed to confirm this hypothesis by determining whether 4-VA has maturation-accelerating effects on male adults in the migratory locust in future” (lines 259-281). (2) We have revised the sentence as “Recently, 4-VA has been identified in the volatiles released by male adults of *S. gregaria* (Torto et al., 2021), whether this volatile has maturation-accelerating effect in this locust species needs further validation” (lines 309-311).